# Analysis of Plant–Fungus Interactions in *Calocybe gambosa* Fairy Rings

**DOI:** 10.3390/plants14182884

**Published:** 2025-09-17

**Authors:** Simone Graziosi, Alessandra Lombini, Federico Puliga, Hillary Righini, Ludovico Dalla Pozza, Veronica Zuffi, Mirco Iotti, Ornella Francioso, Roberta Roberti, Alessandra Zambonelli

**Affiliations:** 1Department of Agricultural and Food Sciences, University of Bologna, Viale G. Fanin 40-44, 40127 Bologna, Bologna, Italy; alessandra.lombini@unibo.it (A.L.); federico.puliga2@unibo.it (F.P.); hillary3001@gmail.com (H.R.); ludovico.dallapozza@studio.unibo.it (L.D.P.); veronica.zuffi3@unibo.it (V.Z.); ornella.francioso@unibo.it (O.F.); roberta.roberti@unibo.it (R.R.); alessandr.zambonelli@unibo.it (A.Z.); 2Department of Life, Health and Environmental Science, University of L’Aquila, Via Vetoio, 67100 Coppito, L’Aquila, Italy; mirco.iotti@univaq.it

**Keywords:** fairy rings, herbaceous plant interactions, plant growth regulation, VOCs

## Abstract

*Calocybe gambosa* (Fr.) Donk is an edible mushroom, highly appreciated especially in Italy. It forms fairy rings (FRs) characterized by a zone of dead vegetation corresponding to the underground-extending mycelial front, followed by a “greener belt” where vegetation is thriving. To better understand this particular phenomenon, the effect of *C. gambosa* mycelium on plants were studied both in situ, across different zones of FRs (external area—EX, fungal front—FF, greener belt—GB, internal area—IN) of three fairy rings, and ex situ on *Poa trivialis* L. Plant community analysis revealed significant changes in plant species composition across the zones, characterized by a decline in diversity and a vegetation shift, from dicotyledons to monocotyledons, progressing from the EX toward the IN, where vegetation gradually begins to reestablish its original composition. Molecular and morphological analyses showed the endophytic colonization of *C. gambosa* mycelium within the herbaceous plants growing at the FF. Ex situ studies indicated pathogenic behavior of *C. gambosa*. After root colonization, it caused growth reduction in *P. trivialis* plants (79% reduction in root length, 76% reduction in leaf length), leaf yellowing, decreased photosynthetic pigments, and root necrosis. The cellulase (endo-1,4-β-glucanase), xylanase, polygalacturonase, and polymethylgalacturonase enzymatic activities of *C. gambosa* support its pathogenic effects. Conversely, volatile organic compounds (VOCs) produced by *C. gambosa* mycelium stimulated shoot development in *P. trivialis* (17% increase in shoot length), which accounts for the formation of the flourishing vegetation zone behind the FF. In contrast, soluble substances produced by *C. gambosa* mycelium did not affect the growth of *P. trivialis*. Our results suggest that *C. gambosa* plays a dual ecological role in regulating plant community dynamics within FRs: it acts as a pathogen by colonizing herbaceous plant roots and, at the same time, promotes vegetation growth through VOC production.

## 1. Introduction

Unusual phenomena caused by the interactions between soil fungi and plants in terrestrial ecosystems are the fairy rings (FRs). These formations can develop both in natural and anthropized environments like managed amenity turfgrasses [1]. FRs are generated by the mycelium of some basidiomycetes that grow radially through soil to form a distinctive rounded structure where basidiomata appear near its outer edge [2,3,4,5,6,7]. The size of FRs can range from a few centimeters to several meters in diameter [1], and the entire zone is characterized by changes in vegetation cover [3].

The mycelium front can affect soil conditions during its extension, interacting with many other organisms and modifying the biotic diversity [8]. Indeed, both the soil physicochemical characteristics [9,10,11,12,13] and the prokaryotic and eukaryotic microbial community often undergo significant changes [8,13,14,15,16,17,18]. Nevertheless, one of the most notable effects occurs on vegetation [6].

FR-forming fungi reduce the diversity of the plant community [8,19] and stimulate and/or suppress the vegetation cover [20,21,22,23]. In many cases, these two opposite effects on vegetation cover occur in the same FR depending on seasonal climate conditions or the position with respect to the fungal front [21,24,25,26].

Based on the effect of the mycelium on vegetation, FRs have been classified by Shantz & Piemeisel [6] into three main types: *type 1* where it is possible to detect a zone of dead vegetation in correspondence with the underground-expanding fungal front followed by a layer of flourishing vegetation [19,25,26]; *type 2* does not include zones of dead vegetation in the proximity of fungal front and it is only visible a layer of flourishing vegetation [26,27]; *type 3* shows no evident effect on vegetation and the FR is only characterized by the presence of basidiomata [26,28]. These positive or negative influences on plant cover seem to be related to the strategy by which the mycelium colonizes the soil to obtain resources. The positive effect shown by FR fungal front on plant cover may be induced by two different processes. The first is due to nutrient release from the fungal hyphae and their increased availability for plants [10,21,24,29,30]. Indeed, an increase in macro and micronutrients like nitric-N [31], phosphate, potassium and magnesium in soil where the mycelium grows has been largely reported in the literature [12,22,23,24,30]. Furthermore, the growing mycelium leads to soil acidification [19,24,32], which, along with the decomposition of plant material, increases the content of organic matter and availability of nutrients, particularly when FRs cause extensive plant die-off [33]. However, studies by Edwards [21] and Gramss et al. [24] found a low correlation between the nutrients released by mycelium of *Agaricus arvensis* Schaeff. and *Marasmius oreades* (Bolton) Fr., and plant growth. The second process involves a phytostimulant effect resulting from the production of hormone-like substances, also known as “fairy chemicals” [34]. They are produced either directly by the fungus or by interaction with other microorganisms [9,35,36]. Notably, *Lepista sordida* (Schumach.) Singer has been shown to produce chemical compounds that mimic auxin-like hormones, promoting the growth in various plant species [35,37,38]. Cao et al. [29] corroborated this effect by demonstrating that *Floccularia luteovirens* (Alb. & Schwein.) Pouzar promotes the growth of grass seedlings through the release of a complex blend of phytostimulant volatile organic compounds (VOCs).

There are many hypotheses about mechanisms causing negative effects on plant cover. At first, the hydrophobic mycelium reduces the soil permeability to water through physical saturation of soil pores and release of hydrophobins [13,24,39]. Moreover, the mycelium can act as a plant pathogen as described for *M. oreades* (Bolton) Fr. [6,39,40]. The mycelium can also alter the beneficial microbial community associated with rhizosphere [19,20,22,23] or accumulate in soil nutrients like ammonium to toxic levels [22,23,41,42]. Another negative effect on plant cover may be due to phytotoxicity, with the release of secondary metabolites such as cyanides, which can be toxic on their own and/or can impede nutrient absorption by plants acting as metal-binding compounds [24,40,43,44].

*Calocybe gambosa* (Fr.) Donk is a basidiomycete that produces *type 1* FRs in grassland and hilly forest ecosystems [13,25,26]. It is of interest not only due to its ecological role as decomposer species but also because of its culinary applications, distinctive organoleptic properties, and traditional harvesting practices in Europe, particularly in Italy [45,46]. The presence of *C. gambosa* in association with herbaceous vegetation significantly influences plant growth. Furthermore, Zotti et al. [13] reported that the mycelium of *C. gambosa* induces modifications in soil physicochemical parameters and affects the composition and dynamics of soil microbial communities. Consistent with other FR-forming basidiomycetes, *C. gambosa* mycelium facilitates the accumulation of soil nutrients required for sustaining its saprotrophic metabolism [13,19,22,23,24,26,30,47,48]. Despite these insights, the trophic interactions between *C. gambosa* and co-occurring herbaceous plants, as well as the direct and indirect effects of its mycelium on plant growth, remain poorly explored.

The aim of this study was to investigate the trophic relationships between *C. gambosa* and herbaceous species associated with its FRs through a novel integrative approach. Specifically, the study intends to: (a) characterize FR formation and associated vegetation in situ; (b) confirm root colonization by *C. gambosa* mycelium both in situ and ex situ by qualitative PCRs with newly designed species-specific primers; (c) assess the direct effects of *C. gambosa* mycelium on the growth of herbaceous plants under controlled ex situ conditions; and (d) evaluate the impact of soluble substances and VOCs produced by the mycelium on herbaceous plant growth. Since the cultivation of *C. gambosa* has not yet been achieved, the results of this study may be relevant for developing future cultivation techniques for this fungus.

## 2. Results

### 2.1. Plant Community Composition of Fairy Rings

We identified 28 plant species from 11 families in the three FRs under observation: seven monocotyledons and 21 dicotyledons; five annual–biennial, and 23 perennial (Appendix A). Diversity indices revealed significant differences in plant communities across the different zones of *C. gambosa* FRs. Species richness (S) and Shannon index (H′) were significantly higher in external area (EX) compared to fungal front (FF), greener belt (GB), and internal area (IN) (*p* < 0.05; Figure 1a,b). In particular, S was highest in the EX, with an average of 23 species, and decreased sharply in the IN (12 species). The H′ also indicated a greater diversity in the EX, with a value of 2.25, compared to the GB, where the lowest diversity was observed (1.90). Regarding evenness (J′), the FF and IN exhibited higher values (0.77 and 0.78, respectively), whereas the GB was characterized by strong dominance of a few plant species (0.67), such as *Anthoxanthum odoratum* L., *Dactylis glomerata* L., and *Festuca rubra* L. However, the statistical difference for J was only found between IN and GB.

The UpSet plot (Figure 2a) revealed marked differences in species composition and sharing among the four FR zones. As previously shown by diversity indices, EXs consistently harbored the highest species richness, with 24 (FR2) or 23 (FR1 and FR3) species, followed by GBs with 18 (GB2 and GB3) and 15 (GB1) species. The FRs with the lowest number of species were FF1, FF3, IN1 and IN3 with 12 species each. Notably, nine species were shared across all zones, six species were uniquely shared among the EXs, and three only between EXs and GBs.

The NMDS ordinations based on Bray–Curtis dissimilarity effectively depicted dissimilarities among plant communities of the four zones of FRs (Figure 2b,c). NMDS analyses showed that EXs are clearly separated along the NMDS1 axis in both ordination plots (Figure 2b,c), reflecting a distinct floristic community despite INs, FFs, and GBs which clustered more closely. Dicotyledons were prevalently found in EXs, while monocotyledons, dominated FFs and GBs (Figure 2b). Annual–biennial species were more dispersed and primarily associated with EXs while perennial species were mostly found in GBs, FFs, and INs. In both ordination plots, the low stress values (0.0) confirm that the two-dimensional configurations provide an accurate representation of ecological dissimilarities in community composition.

The analysis of plant community composition based on relative abundance at the family level (Figure 3a) highlights clear patterns of taxonomic dominance across the four zones. The Poaceae were dominant in all zones, reaching 87.0% abundance in GBs, followed by FFs (82.5%), INs (80.2%), and EXs (66.5%). Ranunculaceae were mostly represented in EXs (11.7%), and with lower percentages in INs (7.0%), FFs (4.8%), and GBs (4.7%). In EX zones, the species of Apiaceae (8.0%), Rubiaceae (5.0%) and Asteraceae (3.8%) also reached the highest abundance, while Fabaceae reached their highest abundance in FFs (2.4%). Boraginaceae, Caryophyllaceae, Lamiaceae, Plantaginaceae, and Polygonaceae were nearly absent across all zones, especially in FFs and INs.

At the species level, many taxa were shared across the different zones of FRs, often displaying similar abundances. However, distinct differences also emerged, with several species exhibiting significant preferences for specific zones, such as the EXs (Figure 3b).

Consistent with earlier family-level analyses, the most abundant species were Poaceae, predominantly concentrated in the GBs, which were heavily dominated by monocotyledonous herbaceous plants. Within GB zones, *D. glomerata* emerged as the most abundant species (25%), followed by *A. odoratum* (22%), *F. rubra* (20%), and *Holcus lanatus* L. (17%), all of which reinforced the strong dominance of perennial monocotyledonous species. Other species of this group were *Lolium perenne* L. (2%), *Poa trivialis* L. (1.7%) and *Poa sylvicola* (1.2%). The dicotyledonous species in the GBs were represented by *Ranunculus bulbosus* L. (4.7%), *Pastinaca sativa* L. (2.1%), *Rumex acetosa* L. (2.0%), *Taraxacum F.H. Wigg.* sp. (1.3%) and the other 4 species with <1% of coverage (Figure 3b). The community composition in FFs and INs was similar to GBs, although a slight increase in dicotyledonous species was registered. Similarly to GBs, *D. glomerata* was the most abundant species in INs, and *Taraxacum* sp. reached its peak abundance (2.5%), while *Trifolium pratense* L. was relatively more abundant in FFs (2.4%).

Dicotyledonous species became more dominant in EXs, with *R. bulbosus, P. sativa* and *C. glabra* that increased to 12.0%, 6.7% and 4.3%, respectively. *Ajuga reptans* L., *Cirsium arvense* (L.) Scop., *Cruciata glabra* (L.) Opiz, *Lathyrus pratensis* L., *Leucanthemum vulgare* Lam., *Salvia pratensis* L., *Tragopogon pratensis* L., *Veronica chamaedrys* L., and *Vicia sativa* L., were recorded only in EXs. A strong reduction was observed in the abundance of grasses such as *D. glomerata* (15%) and *F. rubra* (10%), which dominated GB zones. Some species, including *B. perennis*, *L. perenne*, and *P. trivialis*, were absent in EXs.

### 2.2. Verification of Radical Colonization by C. gambosa Mycelium Through PCR

PCRs using species-specific primers for *C. gambosa* designed in this study (Appendix A), conducted on *A. odoratum*, *D. glomerata*, *F. rubra*, *H. lanatus*, *P. sativa*, *R. bulbosus*, *R. acetosa*, *Taraxacum* sp., and *T. pratense* from the FRs, revealed the presence of *C. gambosa* mycelium within the roots of several samples (Figure 4). Notably, *P. sativa* showed the highest rate of positive detection, with 67% of the root samples positive for *C. gambosa* DNA. Other species exhibiting high positive amplification rates were *A. odoratum* (50%), and *D. glomerata*, *F. rubra*, *H. lanatus*, and *R. acetosa* (33% each). In contrast, species such as *R. bulbosus* and *Taraxacum* sp. showed low levels of detection, with only 17% of samples testing positive. Finally, *T. pratense* showed no positive detection inside the FRs. The control samples collected at least 10 m far from FRs showed no detection of *C. gambosa* mycelium, with the exception of *A. odoratum*, *R. bulbosus*, and *T. pratense*, which showed one positive root sample each out of 27 total analyzed control root segments.

### 2.3. Microscopic Observations of In Situ and Ex Situ Plant Roots

Microscopic observation revealed hyphal colonization in the central root segments of in situ plants, which tested positive by PCR using species-specific primers for *C. gambosa* (Figure 5). Specifically, *C. gambosa* hyphae penetrated the root tissue (Figure 5c), also through root hairs (Figure 5d), and appeared to grow into the apoplast and surround adjacent plant cells (Figure 5g). Moreover, the hyphae appeared capable of crossing the cell wall, and the hyphal coils that conformed to the shape of individual cells supported the assumption of internal colonization (Figure 5e,f).

Similarly, the roots of plants grown in vessels inoculated with *C. gambosa* mycelium in ex situ experiment exhibited comparable root colonization. The mycelium of *C. gambosa* colonized the soil near the roots of *P. trivialis*, forming a conspicuous white mycelial biomass and enveloping the root surface with dense mycelial strands (Figure 5b). Root colonization induced a color change in the root tissue to orange (Figure 5a). Control root samples showed no evidence of mycelial colonization.

Clamp junctions and thin hyphal diameters (<3 µm) were observed in both ex situ and in situ samples (Figure 5g), indicating similar morphological characteristics to *C. gambosa* mycelium in pure culture.

### 2.4. Effects of Active Mycelium on Plant Growth

The mycelium of *C. gambosa* significantly affects the growth of *P. trivialis*, with negative effects both on leaf and root length (Figure 6). The roots and leaves of the inoculated plants were about 79% and 76% shorter than those in control vessels, respectively.

*Poa trivialis* control plants did not show any visible symptoms of infection, whereas those co-cultured with *C. gambosa* showed extensive chlorosis and dieback of the leaves, as well as necrosis of the roots with an infection degree (ID) of 16.75 (Appendix A). In the inoculated vessels, the relative abundances of damage classes 1 to 5 were 5%, 20%, 30%, 25%, and 20%, respectively. The dry weight of the plants co-cultured with *C. gambosa* was significantly lower than that of the control plant, showing a 65% reduction (Table 1).

The control plants showed significantly higher concentrations in all pigments (chlorophyll-a, chlorophyll-b, and carotenoids) than the plants inoculated with *C. gambosa* mycelium (Table 1; Appendix A). Specifically, the average concentrations of chlorophyll-a, chlorophyll-b, and carotenoids in the treated plants decreased by 81%, 70%, and 83%, respectively (Table 1).

The pathogenic activity of *C. gambosa* mycelium on *P. trivialis* was also confirmed by the release of pathogenic enzymes, as determined by enzymatic plate assays (Appendix A). The mycelium of *C. gambosa* produced halos and mycelial areas with the following dimensions: 4.27 ± 1.09 cm^2^ and 3.32 ± 0.14 cm^2^ for endo-1,4-β-glucanase; 1.49 ± 0.18 cm^2^ and 3.06 ± 0.39 cm^2^ for polygalacturonase; 2.21 ± 0.36 cm^2^ and 7.65 ± 0.89 cm^2^ for polymethylgalacturonase; 1.71 ± 0.62 cm^2^ and 14.80 ± 1.80 cm^2^ for xylanase.

### 2.5. Effects of Soluble Substances and VOCs Produced by C. gambosa Mycelium on Plant Growth

The soluble substances produced by *C. gambosa* mycelium in liquid cultures did not result in any significant differences in either leaf or root growth of *P. trivialis* plants. No differences in leaf and root length were observed among plants grown at different liquid medium dilutions or with both MMN and water controls (Appendix A). Instead, the co-culture experiment evaluating the effect of VOCs produced by *C. gambosa* mycelium on shoot elongation of *P. trivialis* seedlings revealed a significant effect (Appendix A). Plants exposed to *C. gambosa* mycelium VOCs exhibited a 17% increase in shoot length (2.77 ± 0.058 mm) compared to the controls (2.37 ± 0.087 mm). It was not possible to evaluate the effect of VOCs on root elongation due to their poor development at the time of measurement.

## 3. Discussion

The complex ecological interactions between fungi and other organisms in FRs have long attracted scientific interest, and several studies have focused on this topic [6,8,13,21,24,39,40,49]. Nevertheless, despite the culinary interest in the FR-forming mushroom *C. gambosa*, only Zotti et al. [13] have examined the soil modifications caused by its mycelial growth and the subsequent effects on microbial communities. The effects of *C. gambosa* on plants, however, have not yet been explored. In our study, the trophic relationship between *C. gambosa* and herbaceous plants was investigated for the first time. An innovative approach combining in situ observations, ex situ experiments simulating natural conditions, and molecular analyses was applied. Specifically, we examined the potential parasitic behavior of *C. gambosa* mycelium and the influence of both soluble substances and VOCs produced by the fungus.

### 3.1. C. gambosa Influences Plant Community Composition Within FRs

Our results showed that the mycelium of *C. gambosa* exerts a significant influence on plant growth and community composition, contributing to the formation of the characteristic greener belt observed just behind the fungal front, where the vegetation is suppressed or senescent. This spatial pattern corresponds to the classical “*type 1*” fairy rings, first documented by Shantz & Piemeisel [6], successively by Hardwick & Heard [31] in turfgrass colonized by *M. oreades*, and further discussed in a recent comprehensive review [26]. Additionally, Fidanza et al. [50] reported a similar zonation effect in managed turfgrass associated with *Agaricus campestris* L.

Pronounced differences in plant community composition were evident among the different zones of FRs (Figure 1, Figure 2 and Figure 3). In particular, the zone external to FR is characterized by a higher diversity and a very different plant community with respect to the internal zones. Biodiversity indices supported these differences while also highlighting specific characteristics of each FR zone. The EX zone, unaffected by the presence of *C. gambosa*, displayed a greater diversity, characterized by a significantly greater species richness and high relative abundances. The IN had the most homogeneous plant community, with low species richness and distributed species abundances. A similar trend was observed in FF. The GB zone differed significantly from EX and was characterized by low evenness, with dominance by a few species. This phenomenon can be attributed to the disturbance caused by the presence of *C. gambosa* mycelium, with the GB zone undergoing the strongest effects of the fungus on both plants and the surrounding environment. The Upset plot and NMDS ordination corroborated these assumptions because EXs from all rings formed a distinct floristic group, while FFs, GBs, and INs clustered separately (Figure 2b,c). Six species occurred only in EXs (Figure 2a), highlighting the filtering effect of fungal activity. This pattern mirrors observations from other fairy-ring-forming fungi, such as *Agaricus arvensis* Schaeff and other *Agaricus* spp. [6,8,19,51]. For instance, Zotti et al. [8] observed an overall decline in plant diversity caused by *A. arvensis* mycelium, suggesting that FRs exert a negative impact on most plant species growing along the fungal path. Moreover, *C. gambosa* induces the formation of GBs with vigorous vegetation composed of a few dominant grass species mainly belonging to Poaceae. Members of this family might be favored by the mycelium activity, and competitive exclusion process can occur towards the other plant families.

In terms of taxonomic shifts, Poaceae showed increasing abundance from the EXs toward the FFs and GBs, supporting the hypothesis that nitrogen mineralization at the fungal front enhances plant productivity [52], favoring perennial, nitrophilous monocotyledonous species (e.g., *Lolium* spp., *Festuca* spp., *Anthoxanthum* spp.). This is further confirmed by the high Ellenberg N-values of these species [53]. Liu et al. [52] showed that the fungal mycelium of *A. campestris*, *Collybia mongolica* (S. Imai) Z.M. He & Zhu L. Yang, and *L. sordida* significantly boost soil ammonium levels and nitrogen availability, reinforcing the nitrophilic response typical of Poaceae. Other grasses, such as *L. perenne* and *P. trivialis*, were absent from EXs, further linking them to the soil zones affected by the activity of *C. gambosa* mycelium. In contrast to *C. gambosa*, other *type 1* FR-forming fungi, such as *A. arvensis*, *A. campestris*, *Agaricus tabularis* Peck, and *M. oreades*, have a detrimental effect on the abundance of perennial species, particularly on the dominant perennial Poaceae. In that case, after the mycelium passage, plant composition underwent a total turnover with a shift towards short-lived, annual species [6,8,19,49]. Why and how different FR-forming fungi favor different plant species remains to be verified. Other families, such as Boraginaceae, Caryophyllaceae, and Lamiaceae, were rare in FFs and INs, suggesting sensitivity of their species to fungal-induced soil changes. As the fungal front moved forward, internal zones may partially recover some floristic complexity, although the legacy effects of mycelial progression remain evident. This suggests that full recovery of the original plant species assemblage may be delayed or incomplete, depending on soil conditions and functional resilience. The long process of restoring the original composition of the plant community has also been observed for other *type 1* FR-forming fungi, particularly *Agaricus* spp. [6,19,21]. Among dicotyledonous species, *R. acetosa* was more abundant in fungal-colonized zones than in EXs, suggesting tolerance or preference for nutrient-rich, disturbed soils. This is consistent with previous findings identifying *R. acetosa* as a plastic species thriving in moist, moderately acidic, nutrient-enriched environments [54,55,56]. In experimental trials, the species demonstrated enhanced root and shoot development under elevated nitrogen and phosphorus conditions, further supporting its affinity for enriched substrates [57]. Similarly, *A. odoratum*, though a monocotyledon, appears well-adapted to fungal-induced disturbances, consistent with its classification as a competitive–stress-tolerant–ruderal (C–S–R) strategy [58,59]. Both species are known for ecological plasticity and colonization of transitional soils [53,60,61]. In contrast, *A. reptans*, *C. arvense*, *C. glabra*, *L. pratensis*, *L. vulgare*, *P. lanceolata*, *S. pratensis*, *T. pratensis*, *V. chamaedrys*, and *V. sativa* were observed exclusively in the EXs, suggesting a loss of competitiveness in soil colonized by *C. gambosa* mycelium.

### 3.2. Direct Pathogenicity of C. gambosa Mycelium on Plants in Fungal Front

The mycelium of *C. gambosa* exhibited two main effects on plants during the expansion of the fungal front: enhanced development of plants forming the greener belt, and had detrimental effects on plants growing in the fungal front (Figure 7).

We showed that negative effects on plant growth of the fungal front are mainly related to a direct parasitic effect of the *C. gambosa* mycelium. *Calocybe gambosa* hyphae were found inside plant roots, both in in situ herbaceous plants and in *P. trivialis* plants grown ex situ together with *C. gambosa* mycelium. The hyphae of *C. gambosa* tended to envelop the root surface, penetrate root tissue, and invade cells. Similar behavior was observed in other fungi that typically develop in grasslands and form fairy rings, such as *M. oreades*, *Lycoperdon curtisii* Berk., and *Holocotylon dermoxanthum* (Vittad.) R.L. Zhao & J.X. Li [62,63]. Although in situ observations revealed that *C. gambosa* colonizes the roots of both monocotyledonous and dicotyledonous species, it occurred more frequently within certain hosts, such as *P. sativa* and *A. odoratum* (Figure 4). The mycelium may exert different effects on these species, as they showed distinct abundances when comparing EXs to other zones. Specifically, *P. sativa* was more common in EXs, suggesting that this plant species is more susceptible to the mycelium of *C. gambosa*. In contrast, *A. odoratum* was more abundant in FFs, GBs, and INs. In this case, the mycelium may be less pathogenic, potentially acting as a harmless endophyte. These findings are supported by the fact that associations between grasses and fungal endophytes are very common, and in some cases, the fungi can act as pathogens [64,65,66]. A similar distribution pattern was observed for *R. acetosa*, which also showed higher abundance in these zones. Interestingly, *C. gambosa* mycelium was also detected in three control root samples taken at least 10 m away from the nearest FR. These samples belonged to *A. odoratum*, *R. bulbosus*, and *T. pratense*. This may indicate an early stage of infection without visible external symptoms, possibly resulting from spore dispersal. Moreover, *C. gambosa* may selectively and partially suppress plant growth at fungal fronts, potentially colonizing not only roots but entire plants, including reproductive organs. This strategy could maintain habitat dominance without causing rapid plant death and may enhance its ability to colonize new areas through vertical transmission via seeds. This mode of transmission, which allows fungal spread via plant reproductive organs, has also been observed in *Cuphophyllus virgineus* (Wulfen) Kovalenko and other fungal species [67,68,69]. The decrease in the dry weight and in the length of both leaves and roots of *P. trivialis* when co-cultured with *C. gambosa* (Figure 6), the damage analyses (Appendix A; Table 1), and the production of plant cell wall-degrading enzymes (Appendix A), such as cellulase, pectinases, and xylanase, by *C. gambosa* mycelium, as observed through enzymatic plate assays, support the pathogenicity of this fungus [70,71]. The halo sizes produced with the same enzymes by several pathogenic fungi, such as *Phaeoacremonium minimum* (Tul. & C. Tul.) Gramaje, L. Mostert and Crous were greater than those obtained with *C. gambosa*. However, when considering the colony diameters of *C. gambosa* and *P. minimum*, normalization of the relative halo dimensions yielded similar values [72]. Moreover, the reduction in chlorophyll and carotenoid content in plants interacting with *C. gambosa* mycelium (Appendix A, Table 1) appears to be related to its pathogenic activity [73,74]. The specific growth strategy of *C. gambosa* may be associated with a highly competitive mechanism for habitat dominance within fungal ecosystems, wherein active mycelia maximize exploitation of available nutrient resources after infected vegetal tissue decomposition. Indeed, the saprophytic activity of *C. gambosa* is demonstrated by its ability to grow under in vitro conditions without plants, as well as in wooded areas in the absence of herbaceous vegetation [75]. Furthermore, similar opportunistic saprophytic capacities were shown by other FR-forming fungi in grasslands [19,22,23,24,30,47,48].

### 3.3. Indirect Effects of C. gambosa Mycelium on Plants Mediated by VOCs in the Greener Belt

The soluble substances excreted by *C. gambosa* mycelium, specifically by the strain Calgam12, did not have any effect on plant growth (Appendix A). That is in contrast with previous studies, which showed that phenoxazine derivatives produced by *C. gambosa* exhibit inhibitory effects on other organisms [76,77]. *Marasmius oreades*, for example, besides parasitic colonization, shows a negative effect on plant growth related to cyanide biosynthesis [39,40]. Other FR-forming fungi, such as *L. sordida*, produce entirely different types of soluble substances which promote plant growth, indicating the existence of diverse strategies involved in fungus–plant interactions [35,37,38].

In our study, although the soluble substances excreted by *C. gambosa* mycelium had no observable effects on plant growth, a growth-promoting effect was demonstrated for the VOCs produced by the mycelium (Appendix A). Specifically, these VOCs enhanced shoot elongation in *P. trivialis*. Based on these results, we hypothesize that this effect may contribute to the observed plant growth promotion in the greener belts of FRs, where the mycelium in the stationary or decline phase produces these VOCs. This positive effect on vegetation could help restore a denser plant cover in areas damaged by the fungus, thereby protecting nearby basidiomata from desiccation or premature fungal predation before spore maturation. These positive effects may act synergistically with increased nutrient availability, particularly NH_4_^+^, NO_3_^−^, P, and K, resulting from the mycelium’s organic matter degradation activity. Additionally, pH reductions observed in previous studies [13] may further support plant nutrient uptake. Moreover, the alleviation of increased soil hydrophobicity associated with active mycelial presence and the restoration of more favorable conditions may also play a role in promoting plant growth [13].

Although fungal VOCs remain less studied than their bacterial counterparts [78], several examples of their plant-growth-promoting properties have been reported. VOCs produced by microfungi, particularly *Trichoderma* spp., are known to stimulate plant growth [79,80,81]. Notably, fungal VOCs can interact with the plant hormonal system; for example, *Cladosporium halotolerans* Zalar, de Hoog & Gunde-Cim. has been shown to upregulate genes involved in gibberellin biosynthesis [82]. Similar effects have also been observed in macrobasidiomycetes. *Floccularia luteovirens*, for instance, has demonstrated growth-promoting activity in alpine meadow plants on the Qinghai–Tibet Plateau. Under in vitro conditions, this fungus enhanced aboveground growth in *Arabidopsis thaliana* (L.) Heynh., induced modifications in root system architecture via the auxin pathway, and regulated plant metabolism [29,83].

## 4. Materials and Methods

### 4.1. Description of Study Area

The study site is situated in the Apennine Mountains, within the Emilia-Romagna region (Montefiorino, Modena, Italy), at an altitude of approximately 800–900 m a.s.l. (Appendix A, Figure 8). The climate is cool temperate (Cf) with summer maximum temperatures often exceeding 30 °C, while winter minima routinely fall below 0 °C. Annual precipitation is over 1500 mm [84]. During the *C. gambosa* fructification period (March–June), the mean precipitation was 9.53 ± 0.76 mm and the mean temperature was 11.64 ± 1.19 °C (Appendix A) [85,86].

### 4.2. In Situ Characterization of Fairy Ring: Study Site and Vegetation Analysis

The FRs developed their characteristic semi-circular shape, with an average radius of around 3 m (Figure 9, Appendix A), and the zones with flourishing vegetation were positioned upstream. In order to confirm that the FRs were formed by *C. gambosa*, the occurrence of its basidiomata was assessed during spring, and their identity was confirmed through morphological [75] and molecular analyses.

Vegetation analysis was conducted along transects laid out across the three selected fairy rings. For each ring, a 250 cm long transect was established and divided into four contiguous semicircular sectors covering all the different zones of the FRs: the internal area (IN), the zone with flourishing vegetation, the greener belt (GB), the fungal front (FF), and the external area (EX) (Figure 9). The cover of each plant species along sectors was assessed during the *C. gambosa* fruiting season (April–May 2024) in their entire areas. The areas of sectors are reported in the Appendix A. Plant species identification was based on the bibliographic repository of Italian flora [88]. Plant cover was measured using the Braun-Blanquet abundance–dominance scale [89,90] and converted into percentages as follows: 5 ≥ 75%; 4 = 51% < x ≤ 75%; 3 = 26% < x ≤ 50%; 2 = 6% < x ≤ 25%; 1 = 1% < x ≤ 5%; + ≤ 1%.

### 4.3. Mycelium Isolation

Twenty 20 *C. gambosa* pure cultures were isolated from basidiomata collected in different habitats of the Apennine mountains between Bologna, Forlì-Cesena and Modena provinces (Appendix A). Three of these strains (Calgam18, Calgam19, and Calgam20) were from the FRs investigated in this study. An additional strain, Calgam12, was isolated from a basidioma collected in an FR dominated by *P. trivialis* (Appendix A), and was selected for further experimental work. Mycelium was isolated on M + P agar medium (pH 6), and the plates were incubated in the dark at 22.5 °C. The M + P medium was prepared by mixing equal volumes of Modified Melin–Norkrans (MMN) [91] and Potato Dextrose Agar (PDA; BD Difco, Franklin Lakes, NJ, USA) [92]. Each isolate was subcultured on M + P medium every month. The basidiomata were subsequently dried and deposited in the herbarium of the “Centro di Micologia” at the University of Bologna (CMI-UNIBO).

### 4.4. Selection of Species-Specific Primers

The internal transcribed spacer (ITS) region of the ribosomal DNA (rDNA) was sequenced for all *C. gambosa* isolates. Amplification was performed using the universal ITS primers ITS1f and ITS4 [93,94], either directly on mycelia by direct PCR [95] or following DNA extraction from basidiomata. PCR conditions are detailed in Appendix A. The resulting amplicons were sequenced using Sanger sequencing at Eurofins Genomics (Köln, Nordrhein-Westfalen, Germany), and the corresponding NCBI accession numbers are listed in Appendix A.

To verify the presence of *C. gambosa* mycelium inside roots of herbaceous plants, species-specific primers for *C. gambosa* (CalgamI and CalgamII; Appendix A) were designed based on both the ITS sequences generated in this study and additional sequences retrieved from GenBank [96] (Appendix A). Sequence alignment was conducted using MUSCLE implemented in MEGA11 [97] to identify conserved regions within the *C. gambosa* ITS sequences suitable for primer targeting. Primer design was carried out using Primer3 [98], and their properties, including melting temperatures and potential primer–primer interactions, were evaluated using OligoAnalyzer version 5 (Integrated DNA Technologies, Coralville, IA, USA). To ensure specificity, the designed primers were aligned with ITS sequences of genetically related species, either generated in this study or obtained from GenBank, to avoid non-target amplification. In vitro validation of primer specificity was performed using DNA extracts from basidiomata of *C. gambosa* and other closely related fungal species, including *Hypsizygus marmoreus* (Peck) H.E. Bigelow (PX218944), *Leucocybe connata* (Schumach.) Vizzini, P. Alvarado, G. Moreno & Consiglio (PX218945), *Lyophyllum deliberatum* (Britzelm.) Kreisel (PX218946), and *Lyophyllum maleolens* M. Melis & Contu (PX218947), which were previously identified by ITS sequencing.

### 4.5. In Situ Detection of C. gambosa Radical Colonization: Plant Sampling and PCRs

For each FR, one plant sample was collected for each of the most abundant herbaceous species (>1% coverage) found in FF and GB zones, including both dicotyledonous and monocotyledonous species. Plants were sampled in the transition zone between FF and GB zones, where it is assumed that the mycelium is more abundant in the roots. Preference was given to specimens grown in close proximity to basidiomata. For each sampled species, a control plant was also collected from the same site, located at least 10 m away from FRs.

All collected plants were taxonomically identified, and their epigeous parts were preserved in the herbarium of the “AQUI, Herbarium Aquilanum” at the University of L’Aquila (Italy). Roots were washed under tap water, and three independent roots were excised from each root system. Portions of the excised roots with evident signs of infection were identified under a stereomicroscope and sectioned into three consecutive 1 cm-long segments. Roots without signs of infection from control plants were processed in the same manner. The central segment was reserved for microscopic observation, while the upstream and downstream segments were used for DNA extraction.

Upstream and downstream segments were surface-sterilized in 70% ethanol for 5 min, followed by a 15 s soak in 0.9% sodium hypochlorite, and finally rinsed three times with sterile water, following the protocol of Cao et al. [99]. To facilitate cell lysis, the root pieces were first added to 300 mL of PL1 buffer of the NucleoSpin^®^ Plant II kit (Macherey-Nagel, Düren, North Rhine–Westphalia, Germany), crushed using a micropestle in sterile sand, and then disrupted with a Tissue Lyser (Qiagen, Venlo, Limburg, The Netherlands) for 10 min at 30 Hz. Then the DNA was extracted with the NucleoSpin^®^ Plant II kit following the manufacturer’s instructions and preserved at −20 °C pending molecular investigations. To verify the presence of *C. gambosa* mycelium inside the roots, nested PCRs were performed on these DNA extracts, first using universal ITS primers, and subsequently using species-specific primers for *C. gambosa* on previous amplicons, under the conditions described above.

### 4.6. Ex Situ Study of C. gambosa Mycelium Direct Effects on Plant Growth

The ability of *C. gambosa* mycelium (strain Calgam12) to influence plant growth was tested on *P. trivialis* because it was the most abundant species in the FR where Calgam12 was isolated, and it was also present in nearly all the studied FRs. Mycelial plugs (0.5 cm in diameter) were aseptically taken from 30-day-old M + P agar cultures of Calgam12 and transferred into 100 mL flasks containing 30 mL of MMN liquid medium (pH 6). These cultures were incubated in the dark at 22.5 °C for two months. Meanwhile, 12 polypropylene vessels with hermetic lids and filters were filled with 250 mL of a soil mixture containing 5 kg, 1 kg, and 1 L of field soil, sand and vermiculite (fragments > 2 mm), respectively. This substrate was sterilized by autoclaving at 121 °C for 1 h on two consecutive days, with overnight cooling at room temperature. Successively, 50 mL of MMN (pH 6) was added to each vessel and autoclaved again for 20 min.

Following sterilization, six vessels were inoculated with the *C. gambosa* liquid cultures. The exhausted liquid medium was removed, and the mycelium was washed with sterile distilled water before being placed directly onto the soil mixture. The inoculum was gently covered with surrounding soil using a sterile loop to prevent dehydration. The remaining six vessels were left uninoculated and served as controls. All vessels were incubated in the dark at 22.5 °C until full mycelial colonization, which occurred after approximately three months. At this stage, 0.3 g of *P. trivialis* seeds, provided by “Seed Research and Testing Laboratory”, at the University of Bologna (LaRAS-UNIBO), were sown into each vessel. Seeds were first soaked in 500 mL of sterile distilled water with two drops of Tween 20, and then surface sterilized by immersion in 35% hydrogen peroxide for 90 min, and finally rinsed three times with sterile water. The vessels were subsequently transferred to a greenhouse and grown at 22.5 °C.

Plants from six vessels (three inoculated and three controls) were carefully removed one month after sowing to observe root colonization. The shoot and root lengths of 25 individual plants per vessel (five plants for each vertex of a pentagon, 3 cm per side, placed in the center of the vessel) were measured using a ruler. Three months after sowing, plants from the remaining vessels (three inoculated and three controls) were harvested and pooled. Leaf and root symptoms were visually assessed and categorized using a damage scale ranging from 0 to 5 (Appendix A): 0, absence or slight traces of symptoms; 1, leaf yellowing or root necrosis affecting less than 5% of the total surfaces, respectively; 2, 5–10% of leaf and root surfaces with symptoms; 3, 25–50% of the leaf and root surfaces with symptoms; 4, 50–75% leaf and root surfaces with symptoms and partial leaf desiccation; 5, 75–100% of leaf and root surfaces with symptoms and leaves extensively desiccated. In cases where leaf yellowing and root necrosis in the same plant fell into different categories, the higher of the two scores was recorded.

The percentage abundance of each damage class was determined along with the dry biomass of plants from both inoculated and control vessels. Disease severity (infection degree, *ID*) was computed using a scale of total number of classes with the Townsend-Heuberger formula [100,101,102]:(1)$$ID\;\left(\%\right)=\;\frac{\sum_{i=1}^{n}\left(n_{i}\times v_{i}\right)}{N\;\times V}$$
where *n_i_* is the number in one class, *v_i_* is the damage class, *N* is the total number, *V* is the highest class, and *i* is the number of classes.

The carotenoid and chlorophyll content of herbaceous plant leaves was determined spectrophotometrically according to Righini et al. [103]. Three 20 mg samples of leaves, located at the vertices of an equilateral triangle (3 cm per side), placed in the center of the vessel, were taken by each of the six vessels (three inoculated and three controls) after three months of growth. The leaf samples were soaked in 1 mL of methanol, crushed using micro-pestels, and left in methanol for 24 h. Then, the chlorophyll-a, chlorophyll-b and carotenoid contents were obtained by measuring the absorbance at 665, 652 and 470 nm wavelengths, respectively.

### 4.7. Ex Situ Mycelial Enzyme Production

To estimate the pathogenicity of *C. gambosa*, specific enzymatic assays were conducted in 9 cm Petri dishes containing 25 mL of different agarized media. Cellulase (endo-1,4-*β*-glucanase) and xylanase activity were assessed on PYE medium (peptone 0.5 g L^−1^, yeast extract 0.1 g L^−1^, agar 16 g L^−1^) supplemented with 0.5% sodium carboxymethylcellulose (Sigma-Aldrich, St. Louis, MO, USA) for cellulase activity, and a minimal medium containing 0.3% NaNO_3_, 0.1% KH_2_PO_4_, 0.05% MgSO_4_, 0.1% yeast extract, and 1.2% agar, supplemented with 0.5% beechwood xylan (Sigma-Aldrich, St. Louis, MO, USA) for xylanase activity [104,105]. Six Petri dishes were prepared for each enzymatic assay, inoculating them with 0.5 cm mycelial plugs taken from 30-day-old M + P (pH 6) cultures of Calgam12. Petri dishes were subsequently incubated at 22.5 °C in the dark for one month. The cellulase activity of Calgam12 was visualized by staining Petri dishes with a 0.2% Congo red solution for 30 min with agitation at room temperature. The plates were then rinsed with distilled water, washed with 1 M NaCl for 30 min with agitation and rinsed again with distilled water [106]. The xylanase activity was detected by observing the halos obtained by staining the inoculated plates with a 0.2% Congo red water solution for 15 min, then destained with 1 M NaCl [72].

Pectinase activity, specifically polygalacturonase and polymethylgalacturonase, was assessed using plates containing a culture medium after Eriksson & Pettersson [107]. This medium was supplemented with 2% agar and 0.5% sodium salt of polygalacturonic acid (Sigma-Aldrich, St. Louis, MO, USA) derived from citrus fruit for polygalacturonase activity and 0.5% citrus pectin (Sigma-Aldrich, St. Louis, MO, USA) for polymethylgalacturonase activity. The growth media were sterilized according to the protocol described by Ayers et al. [108] for polygalacturonase and by Durrands & Cooper [109] for polymethylgalacturonase. Following sterilization, the pH was adjusted to 5 for polygalacturonase and 8 for polymethylgalacturonase using 1 M NaOH and 1 M HCl, respectively. Six replicates were prepared for each of the two enzymatic assays. Petri dishes were subsequently incubated at 22.5 °C in the dark for two months, since the mycelium of *C. gambosa* grows slowly on these media. The halo due to the enzyme activity was visualized by incubating the plates at 25 °C for 30 min, then flooding them with a 1% cetylmethylammonium bromide (CTAB) solution in distilled water for 5 min, and heating them to 30 °C for 30 min [110].

For each enzymatic test, colony growth and enzyme activity were assessed by measuring the area covered by the mycelium and the area of the halos surrounding the colonies in their respective medium. The areas (cm^2^) were calculated according to Graziosi et al. [111], assuming an elliptical shape covered by the mycelium or its halo, as reported by Tryfinopoulou et al. [112] with the following formula:(2)$$A=R_{1}\times R_{2\;}\times \pi$$
where *A* is the fungal colony area (cm^2^) and *R*_1_ and *R*_2_ are the two perpendicular radii, respectively.

### 4.8. Microscope Observations of Ex Situ and In Situ C. gambosa Root Colonization

Microscopic observations of in situ plants were conducted on central root segments excised from roots that showed positive species-specific PCR amplification of *C. gambosa* DNA. Root samples from ex situ plants were selected from three inoculated and three control plants (three root samples per plant, 18 in total). The plants were collected at the vertices of an equilateral triangle (3 cm per side) and placed in the center of the vessel. The root samples were rinsed under tap water to remove soil debris prior to microscopic observation.

Microscopic analysis, on both ex situ and in situ root samples, was performed using a Nikon Eclipse TE2000 U Inverted Microscope (Nikon Corporation, Tokyo, Kantō, Japan), and images were captured with a Nikon DS-Fi3 digital camera (Nikon Corporation, Tokyo, Kantō, Japan) to assess fungal hyphal colonization. Before observation, the roots were stained following a modified protocol after Barrow & Aaltonen [113]. Root segments were placed in 2 mL tubes containing 1.8 mL of 10% KOH solution and incubated at 120 °C for 30 min to bleach the tissue. After bleaching, the solution was discarded, and the samples were rinsed three times with distilled water. A staining solution (1.8 mL) composed of 5% trypan blue and 5% glacial acetic acid was then added, and the tubes were incubated at 90 °C for 10 min. Finally, root segments were rinsed with distilled water and mounted on microscope slides with a drop of lactic acid for observation.

### 4.9. Ex Situ Influence of Soluble Substances Produced by the C. gambosa Mycelium on Plant Growth

The effects of soluble substances produced by *C. gambosa* mycelium were evaluated by growing plants in the presence of different concentrations of filtered liquid culture media. A total of 24 liquid cultures of *C. gambosa* mycelium (strain Calgam12) were grown in 30 mL of MMN as previously described. Additionally, 32 polypropylene vessels, each fitted with hermetic lids and filters and filled with the standard soil mixture, were prepared. The vessels were sterilized by autoclaving at 121 °C for 1 h on two consecutive days, with overnight cooling at room temperature between sterilization cycles.

The liquid cultures were first passed through paper filters, then filtered using 0.2 μm Millipore^®^ membrane filters. Serial dilutions of the filtered cultures (100%, 75%, 50%, and 25%) were prepared using sterile distilled water. For each concentration, 50 mL of the solution was added to six vessels. Two control treatments were included: four vessels received 50 mL of fresh MMN medium (100% concentration), and another four received 50 mL of sterile distilled water. In all vessels, 0.3 g of *P. trivialis* seeds, surface-sterilized as previously described, were sown directly into the substrate. All vessels were maintained in a greenhouse at a temperature of 22.5 °C. One month after sowing, shoot and root lengths were measured, using a ruler, on 25 plants per vessel, five plants for each vertex of a pentagon (3 cm per side), placed in the center of the vessel.

### 4.10. Ex Situ Influence of VOCs Produced by the C. gambosa Mycelium on Plant Growth

The effect of VOCs produced by *C. gambosa* mycelium was evaluated using a plate co-culture system. Mycelial plugs (0.5 cm in diameter) from 30-day-old M + P agar cultures (pH 6) of the Calgam12 strain were aseptically transferred into six Petri dishes, each containing 25 mL of M + P medium (pH 6). These cultures were incubated in the dark at 22.5 °C for two months to allow mycelial development.

Following mycelial growth, six additional Petri dishes were prepared, each containing a sterile filter paper disc (8 cm in diameter) moistened with 1.4 mL of sterile distilled water. Onto each disc, 0.1 g of *P. trivialis* seeds, previously surface-sterilized according to the protocol previously described, were evenly sown. The lids of both sets of plates were removed, and the plates containing mycelium were inverted and placed over the seed-containing plates. The plate pairs were then sealed using three layers of parafilm and an additional layer of insulating tape to minimize VOC loss. Six control plate pairs were assembled following the same procedure but without the presence of fungal mycelium. After two weeks of incubation, shoot elongation was measured, using a ruler, for 25 plantlets per plate pair, five plants for each vertex of an equilateral pentagon (3 cm per side), placed in the center of the plate.

### 4.11. Statistical Analysis

The statistical analyses were carried out using R Studio 2024.09.1+394. Significant differences between treatment tests (average concentrations of chlorophyll-a, chlorophyll-b, and carotenoids, average weight of dry plants, biodiversity indexes, leaf and root length of plants) were tested by one-way ANOVA, and the means were compared by Tukey’s test (*p* ≤ 0.05). Alpha and beta diversity within and among sites were analyzed using biodiversity indices: species richness (S), Shannon index (H′), and Pielou’s evenness (J′). An upset was used to highlight intersecting species groups among FR zones. Non-metric multidimensional scaling (NMDS) based on Bray–Curtis dissimilarity was used to assess variation in species composition across FRs, considering the morphological type (monocotyledons vs. dicotyledons) and the life form (annuals–biennials vs. perennials). The relative abundance of plant families and species among FR zones was investigated using a stacked bar chart and a heatmap, respectively. In the heatmap, variables were reordered according to the results of Bray–Curtis dissimilarity matrix.

## 5. Conclusions

Our results indicate that *C. gambosa* plays a dual ecological role in regulating plant community dynamics within its FRs. On one hand, the active mycelial front exhibits parasitic-like behavior by suppressing plant growth and reducing competition in its immediate zone of expansion. On the other hand, the greener belt formed just behind this front appears to benefit from nutrient enrichment and the release of VOCs, promoting enhanced vegetation growth. These findings suggest that *C. gambosa* not only modifies soil chemistry but also actively shapes aboveground biodiversity through complex biotic interactions. Further exploration of its bioactive compounds could open promising applications in sustainable agriculture and biostimulants. Nevertheless, these applications need to be verified at the field scale through specific trials. Moreover, interactions with herbaceous plants may represent an essential phase in the life cycle of *C. gambosa*, playing a role in the fructification process. This insight could open new opportunities for the cultivation of this valuable mushroom.

## Figures and Tables

**Figure 1 plants-14-02884-f001:**
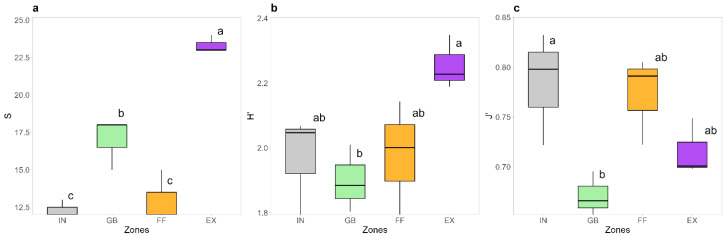
Boxplots showing the distribution of diversity indices in the internal area (IN), greener belt (GB), fungal front (FF), and external area (EX) across fairy rings of *Calocybe gambosa* (FR1, FR2, FR3): (**a**) S—species richness; (**b**) H′—Shannon index; (**c**) J′—Pielou’s evenness. Different letters indicate statistically significant differences according to Tukey’s test (*p* < 0.05).

**Figure 2 plants-14-02884-f002:**
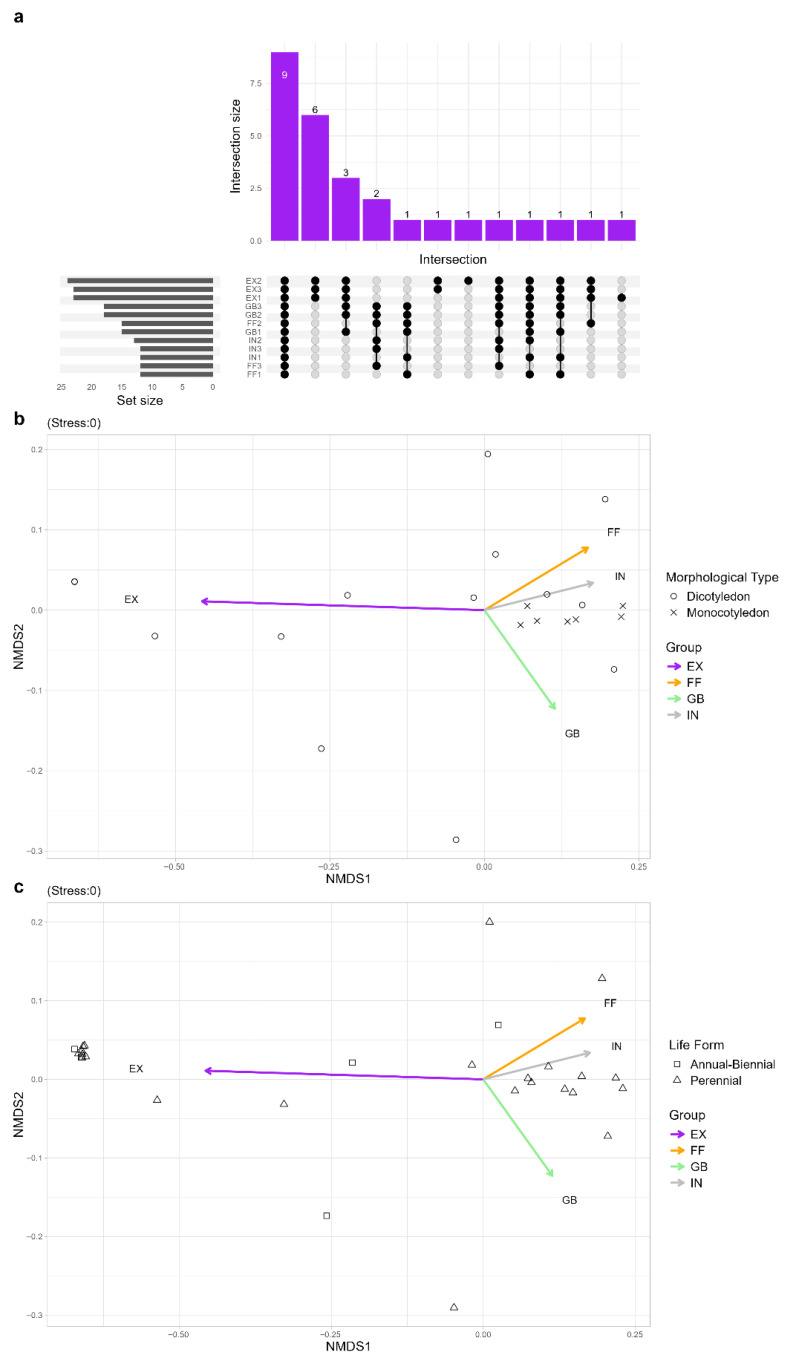
Comparison of species composition between internal areas (INs), greener belts (GBs), fungal fronts (FFs), and external areas (EXs) based on morphological and ecological traits: (**a**) Upset plot showing the number of shared and unique species between zones. The intersection bars represent the number of species shared among zones of the three FRs, while the set size bars indicate the total number of species. (**b**,**c**) Non-metric multidimensional scaling (NMDS) ordination based on species composition, with samples grouped by zones. Arrows indicate the direction and strength of group separation. (**b**) NMDS ordination with species categorized by morphological type: monocotyledons (×) and dicotyledons (○). (**c**) NMDS ordination with species categorized by life form: annual–biennial (□) and perennial (△). The stress value is reported above each NMDS.

**Figure 3 plants-14-02884-f003:**
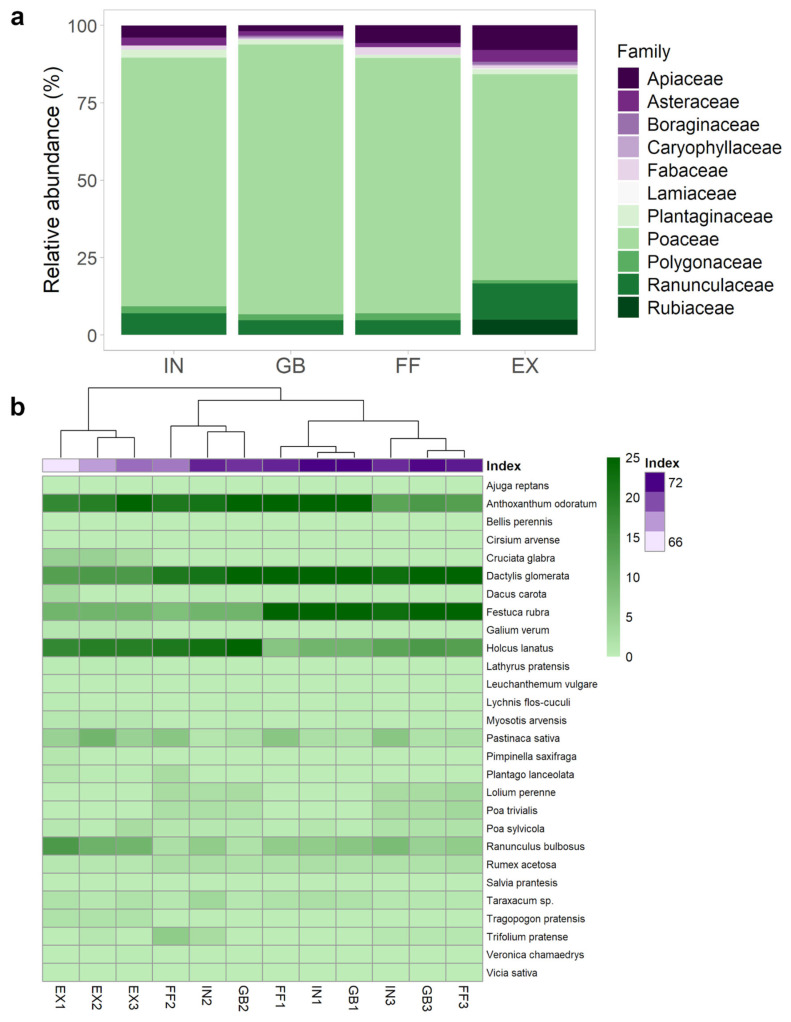
(**a**) Stacked bar chart showing the mean relative abundance (%) of dominant plant families across the four concentric zones, internal areas (INs), greener belts (GBs), fungal fronts (FFs), and external areas (EXs), from the three fairy rings (FR1, FR2, FR3). (**b**) Heatmap of the relative abundance of herbaceous species recorded in the four zones of the three fairy rings. Darker green shades indicate higher relative abundances of individual species across sampling plots. The top dendrogram represents species-level clustering based on Bray–Curtis dissimilarity, where darker purple shades indicate stronger ecological association between species groups.

**Figure 4 plants-14-02884-f004:**
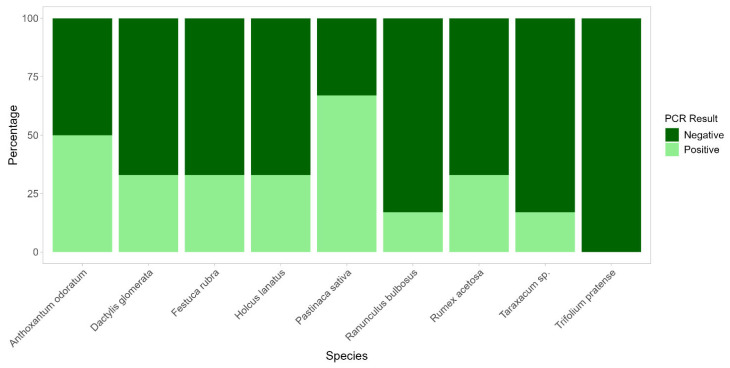
Species-specific PCR detections of *Calocybe gambosa* mycelium in root samples of in situ herbaceous species from fairy rings (FRs). For each species, the percentages of positive (light green) and negative (dark green) root samples were reported following nested PCR with *C. gambosa* species-specific primers (CalgamI-CalgamII).

**Figure 5 plants-14-02884-f005:**
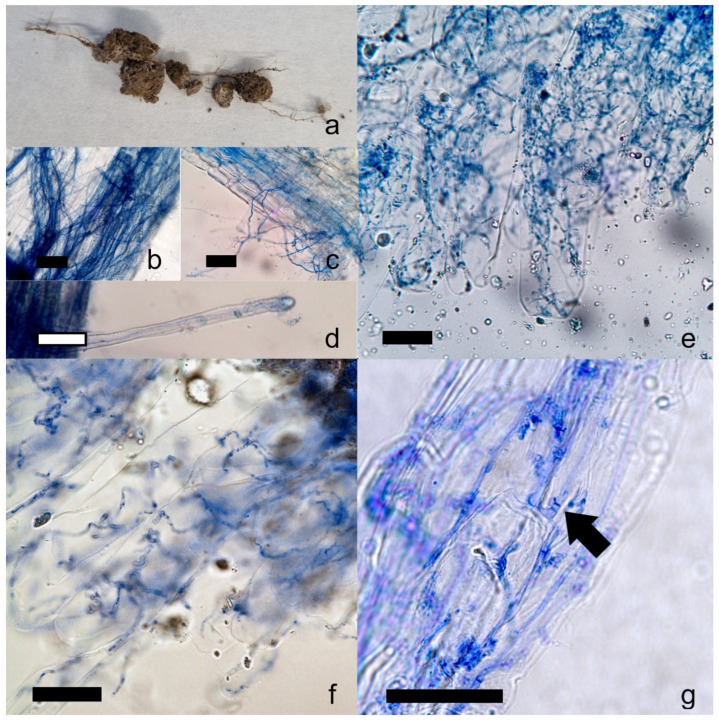
Colonization of the *Poa trivialis* roots by the mycelium of *Calocybe gambosa*. The mycelium exhibited the same behavior in natural samples (**a**–**e**) and semi-natural samples (**f**,**g**) from co-culture tests. The bars correspond to 10 μm. The dark arrow indicates a clump junction. Hyphae were coloured with a trypan blue staining solution.

**Figure 6 plants-14-02884-f006:**
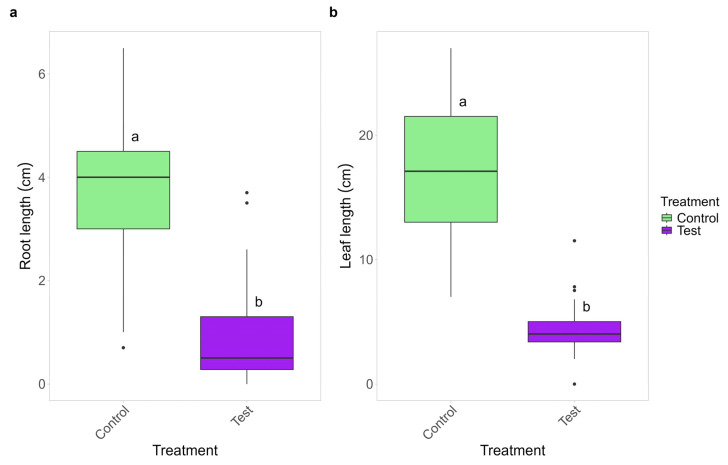
Ex situ co-culture test of *Calocybe gambosa* and *Poa trivialis* effect on elongation of roots (**a**) and leaves (**b**). Different letters indicate a difference between treatments according to Tukey’s test (*p* < 0.05).

**Figure 7 plants-14-02884-f007:**
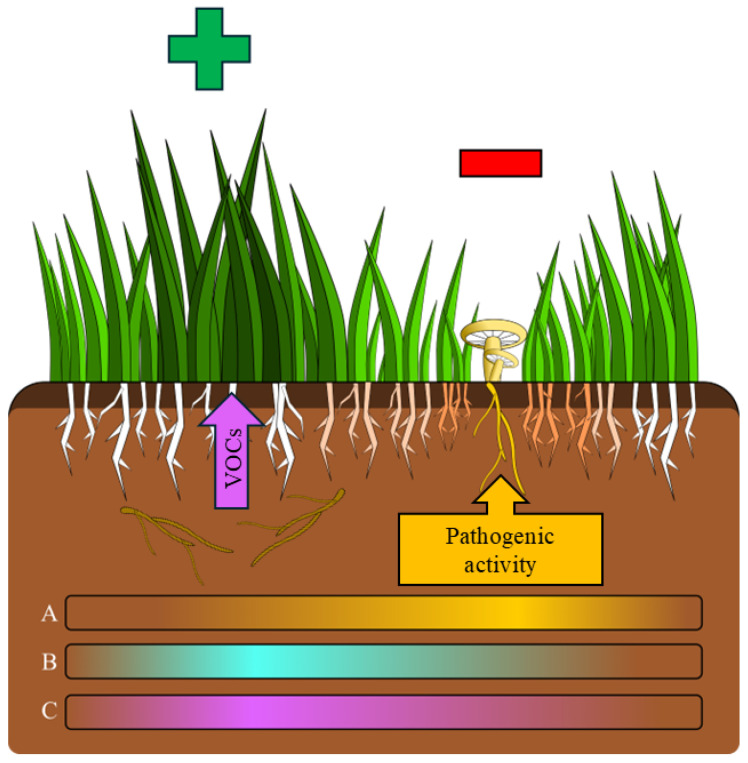
Model of interactions between *Calocybe gambosa* mycelium and plants within fairy rings (FRs). The intensity of the shading in the bars at the bottom of the figure indicates the relative strength of the analyzed factors: (A) mycelial activity, (B) pH decrease and nutrient availability, (C) volatile organic compound (VOC) action. The mycelium reaches its maximum activity at the fungal front, where it colonizes plant roots and exhibits pathogenic behavior (−). After the mycelium advances, the decrease in pH and increase in nutrient availability, along with the loss of fungal detrimental activity and the release of VOCs from mycelium in the stationary or decline phase, stimulate plant growth (+).

**Figure 8 plants-14-02884-f008:**
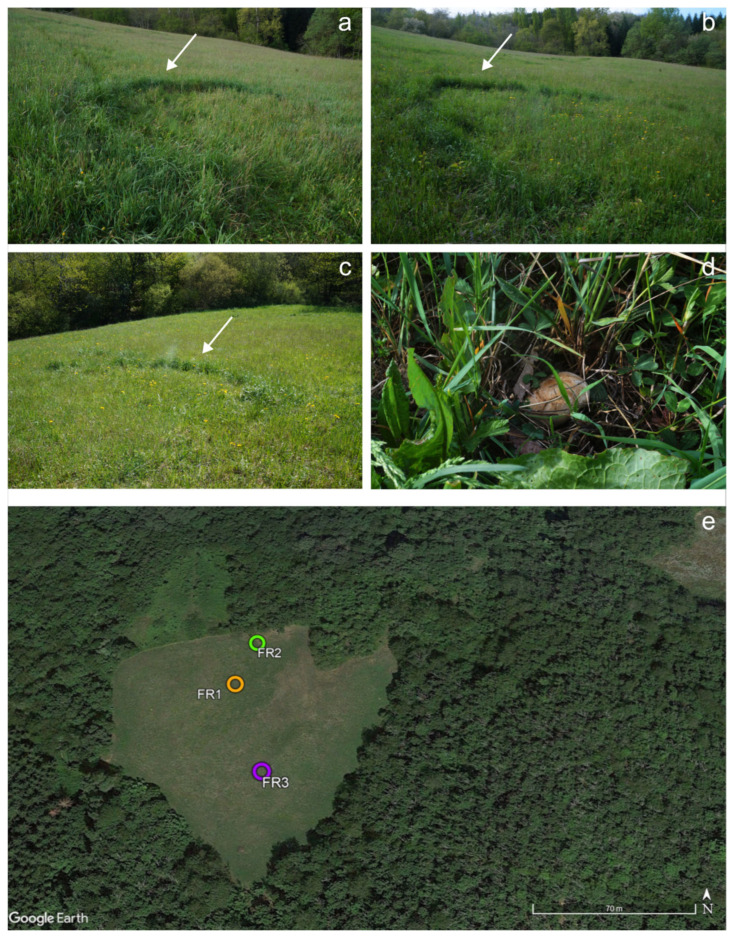
(**a**–**c**) *Calocybe gambosa* fairy rings (FRs; FR1, FR2, FR3) conformations at the productive site. The FRs assumed the typical semi-circular shape. The white arrows indicate the greener belt of FRs. (**d**) Basidioma of *C. gambosa* found in the FR2, (**e**) spatial disposition of the three fairy rings in the grassland Montefiorino (Modena, Emilia Romagna, Italy) site [87].

**Figure 9 plants-14-02884-f009:**
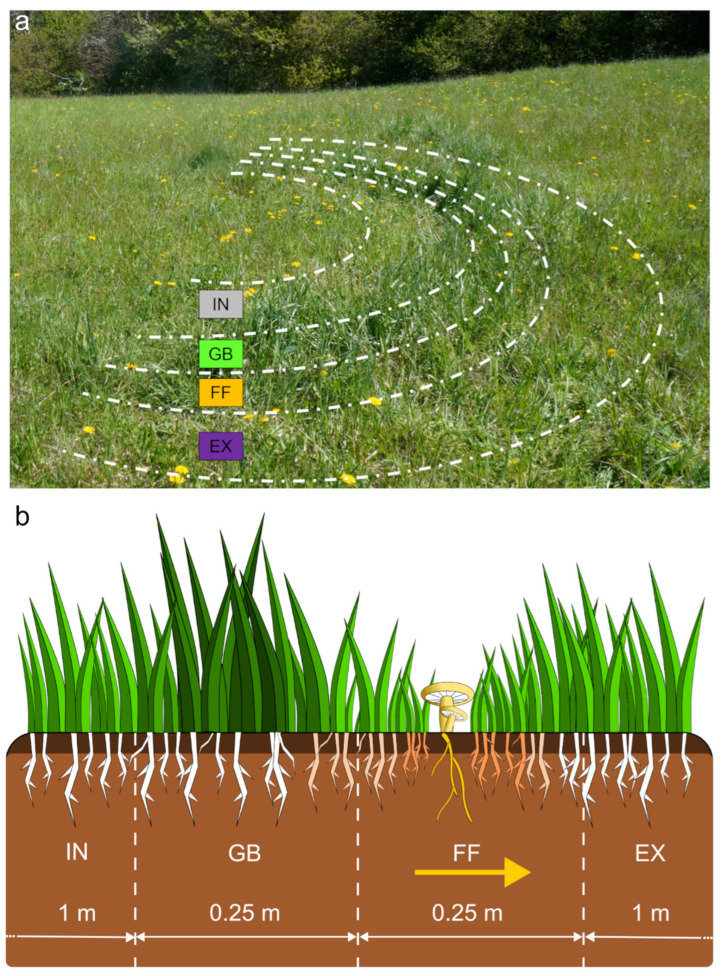
Schematic representation of the different zones analyzed in fairy rings (FRs): internal area (IN), greener belt (GB), fungal front (FF), and external area (EX). (**a**) Dashed lines represent the boundaries of different FR zones; (**b**) white arrows indicate the average length of the zones, while the yellow arrow represents the direction of mycelium growth.

**Table 1 plants-14-02884-t001:** Ex situ assessment of damage to *Poa trivialis* caused by *Calocybe gambosa* mycelium after three months of co-culture under semi-natural conditions. The comparison of the mean values between control and co-cultured plants is reported.

Treatment	Average Weight of Dry Plant (g)	Chlorophyll-a (µg/mg)	Chlorophyll-b (µg/mg)	Carotenoid (µg/mg)
Control	0.069 ± 0.0084 a	1.17 ± 0.060 a	0.57 ± 0.023 a	0.18 ± 0.0079 a
Co-cultured	0.024 ± 0.0012 b	0.22 ± 0.017 b	0.17 ± 0.017 b	0.031 ± 0.0073 b

Note: All data are reported as mean ± standard error (SE). Within columns, different letters indicate a difference between treatments according to Tukey’s test (*p* < 0.05).

## Data Availability

All data generated or analyzed during this study are included in this published article and its Appendix A. ITS sequences of *C. gambosa* strains were deposited at NCBI (GenBank) with the following accession numbers: PV628678–PV628696.

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
