# Peer review of "Analysis of Plant–Fungus Interactions in Calocybe gambosa Fairy Rings"

_plants, 2025, doi:10.3390/plants14182884_

Round 1

Reviewer 1 Report

Comments and Suggestions for Authors

In this manuscript, Simone Graziosi and colleagues analyzed the effect of C. gambosaCalocybe gambosa (Fr.) Donk mycelium on plants. I have following comments:

1, For the Title, I suggest to employ “Analaysis of Plant-Fungus Interactions in Calocybe gambosa Fairy Rings”.

2, For the Abstract section, more values should be presented. For instance, authors stated that “volatile organic compounds (VOCs) produced by C. gambosa mycelium stimulated shoot development in P. trivialis”, but the values were not provided.

3, For the introduction section, practical interests of this study should be stated in the last paragraph.

4, Genotypes of plant and microbe examined in this study should be described in the Materials

5, Methods for random sampling should be introduced in the Methods.

6, For the results, Figure S1 is very important and should be placed in the main Figures

7, A Figure depicting the model proposed by the authors is appealing.

Author Response

Reviewer 1

1, For the Title, I suggest to employ “Analaysis of Plant-Fungus Interactions in Calocybe gambosa Fairy Rings”.

2, For the Abstract section, more values should be presented. For instance, authors stated that “volatile organic compounds (VOCs) produced by C. gambosa mycelium stimulated shoot development in P. trivialis”, but the values were not provided.

3, For the introduction section, practical interests of this study should be stated in the last paragraph.

4, Genotypes of plant and microbe examined in this study should be described in the Materials

5, Methods for random sampling should be introduced in the Methods.

6, For the results, Figure S1 is very important and should be placed in the main Figures

7, A Figure depicting the model proposed by the authors is appealing.

RESPONSE

Comments 1: For the Title, I suggest to employ “Analaysis of Plant-Fungus Interactions in Calocybe gambosa Fairy Rings”.

Response 1: Thank you for the suggestion. We have updated the title. Lines 2−3.

Comments 2: For the Abstract section, more values should be presented. For instance, authors stated that “volatile organic compounds (VOCs) produced by C. gambosa mycelium stimulated shoot development in P. trivialis”, but the values were not provided.

Response 2: Thank you for your suggestions. We have added more data in the abstract. Lines 25−26, 30−31.

Comments 3: For the introduction section, practical interests of this study should be stated in the last paragraph.

Response 3: Thank you for your suggestions. We have added a sentence about that in the introduction. Lines 112−114.

Comments 4: Genotypes of plant and microbe examined in this study should be described in the Materials

Response 4: Thank you for your suggestions. The accession numbers of the ITS sequences used in primer design are provided in Supplementary Tables S4 and S6 of the Supplementary Files. We have also added the accession numbers of the fungal specimens employed for the specificity test during the design of the C. gambosa species-specific primers. Lines 567–570. Furthermore, we have included the origin of the P. trivialis seeds used in the experiments. Lines 620–621.

Comments 5: Methods for random sampling should be introduced in the Methods.

Response 5: Thank you for your suggestions. Regarding the section “In situ detection of C. gambosa radical colonization: plant sampling and PCRs”, we have already specified: “Plants were sampled in the transition zone between FF and GB zones, where it is assumed that the mycelium is more abundant in the roots. Preference was given to specimens growing in close proximity to basidiomata.”. This was the strategy we used to select the plants in the in situ area. Lines 576–580. We have also added information about the samplings conducted in the ex situ experiments. Lines 628−629, 649−650, 702–703, 736–737, 754−756.

Comments 6: For the results, Figure S1 is very important and should be placed in the main Figures.

Response 6: Thank you for your suggestions. We have moved Figure S1 into the main text and have accordingly updated the figure numbers. Line 218.

Comments 7: A Figure depicting the model proposed by the authors is appealing.

Response 7: Thank you for your suggestions. We have added a new figure representing the model proposed in the discussion. Line 397.

Reviewer 2 Report

Comments and Suggestions for Authors

The manuscript “Plant-Fungus Interactions in Calocybe gambosa Fairy Rings” provides a thorough investigation of how C. gambosa interacts with herbaceous plants. Using vegetation surveys, molecular assays, ex situ inoculation, microscopy, enzyme tests, and VOC bioassays, the study shows that C. gambosa has a dual role, pathogenic at the fungal front but growth-promoting in the greener belt through VOCs and nutrient cycling. These findings advance understanding of fairy ring ecology and suggest potential applications in turfgrass, pasture, and agriculture.

Minor Revisions Suggested

    • Clarify if primers can also be applied to environmental DNA (eDNA) for mapping fungal presence.
    • Strengthen comparisons with other FR-forming fungi (e.g., Marasmius oreades, Lepista sordida, Agaricus spp.).
    • Expand on the ecological nuance between pathogenicity and conditional endophytism (e.g., differences in A. odoratum vs. P. sativa colonization).
    • Comment on the biological/ecological relevance of the relatively modest VOC-induced shoot elongation at the community scale.
    • For enzymatic assays, contextualize halo sizes relative to known plant pathogens and clarify how replicate variability was addressed statistically.
    • While the agricultural/biostimulant potential of VOCs is intriguing, this claim should be moderated by acknowledging the need for field-scale validation.

Questions for Authors

  1. Were root colonization levels consistent across plant species, or was there evidence of host preference (e.g., P. sativa vs. A. odoratum)?
  2. Beyond Poa trivialis, do the VOC effects extend to other grasses dominant in GB zones (e.g., Dactylis glomerata, Festuca rubra)?
  3. Do you consider C. gambosa to act primarily as a necrotroph, or could it initially establish as a biotroph/endophyte before switching to pathogenic behavior?
  4. Could soil nutrient conditions (e.g., nitrogen enrichment) modulate the balance between pathogenic suppression at the fungal front and growth promotion behind it?
  5. Were soil physicochemical properties (pH, N, P, K) considered in explaining NMDS community patterns, or are these attributed solely to fungal activity?
  6. Do you hypothesize that plant colonization is essential for triggering C. gambosa fruiting, as suggested in the conclusions?

Author Response

Reviewer 2

The manuscript “Plant-Fungus Interactions in Calocybe gambosa Fairy Rings” provides a thorough investigation of how C. gambosa interacts with herbaceous plants. Using vegetation surveys, molecular assays, ex situ inoculation, microscopy, enzyme tests, and VOC bioassays, the study shows that C. gambosa has a dual role, pathogenic at the fungal front but growth-promoting in the greener belt through VOCs and nutrient cycling. These findings advance understanding of fairy ring ecology and suggest potential applications in turfgrass, pasture, and agriculture.

Minor Revisions Suggested

  • Clarify if primers can also be applied to environmental DNA (eDNA) for mapping fungal presence.
  • Strengthen comparisons with other FR-forming fungi (e.g., Marasmius oreades, Lepista sordida, Agaricus spp.).
  • Expand on the ecological nuance between pathogenicity and conditional endophytism (e.g., differences in odoratumvs. P. sativa colonization).
  • Comment on the biological/ecological relevance of the relatively modest VOC-induced shoot elongation at the community scale.
  • For enzymatic assays, contextualize halo sizes relative to known plant pathogens and clarify how replicate variability was addressed statistically.
  • While the agricultural/biostimulant potential of VOCs is intriguing, this claim should be moderated by acknowledging the need for field-scale validation.

Questions for Authors

  1. Were root colonization levels consistent across plant species, or was there evidence of host preference (e.g., P. sativa vs. A. odoratum)?
  2. Beyond Poa trivialis, do the VOC effects extend to other grasses dominant in GB zones (e.g., Dactylis glomerata, Festuca rubra)?
  3. Do you consider C. gambosa to act primarily as a necrotroph, or could it initially establish as a biotroph/endophyte before switching to pathogenic behavior?
  4. Could soil nutrient conditions (e.g., nitrogen enrichment) modulate the balance between pathogenic suppression at the fungal front and growth promotion behind it?
  5. Were soil physicochemical properties (pH, N, P, K) considered in explaining NMDS community patterns, or are these attributed solely to fungal activity?
  6. Do you hypothesize that plant colonization is essential for triggering C. gambosa fruiting, as suggested in the conclusions?

RESPONSE

Minor Revisions Suggested

Comments 1: Clarify if primers can also be applied to environmental DNA (eDNA) for mapping fungal presence.

Response1: Thank you for your suggestions. Yes, the primers are sufficiently specific to be used on eDNA samples. In a parallel investigation, we applied these primers to soil DNA extracts. However, since this application was not part of the current manuscript and is not directly relevant to our results, we chose not to include it in the text.

Comments 2: Strengthen comparisons with other FR-forming fungi (e.g., Marasmius oreades, Lepista sordida, Agaricus spp.).

Response 2: Thank you for your suggestions. We have added new sentences to strengthen the comparison with other FR-forming fungi. Lines (335−337, 351−356, 362−364).

Comment 3: Expand on the ecological nuance between pathogenicity and conditional endophytism (e.g., differences in A. odoratum vs. P. sativa colonization).

Response 3: Thank you for your suggestions. We have added several sentences to expand the ecological nuance between pathogenicity and conditional endophytism. Lines 423−425.

Comments 4: Comment on the biological/ecological relevance of the relatively modest VOC-induced shoot elongation at the community scale.

Response 4: Thank you for your suggestions. We have added several sentences commenting on the biological and ecological relevance of the relatively modest VOC-induced shoot elongation at the community scale. Lines 470−474.

Comments 5: For enzymatic assays, contextualize halo sizes relative to known plant pathogens and clarify how replicate variability was addressed statistically.

Response 5: Thank you for your suggestions. We have added several sentences contextualizing halo sizes relative to known plant pathogens using the same plate assay. Lines 441−445. The variability of the replicates was addressed using the standard error (SE). The values are reported in Supplementary Table S2, as indicated in the note: “The halo area and mycelial area are reported as mean ± standard error (SE)”.

Comments 6: While the agricultural/biostimulant potential of VOCs is intriguing, this claim should be moderated by acknowledging the need for field-scale validation.

Response 6: We have moderated the claim about the agricultural/biostimulant potential of VOCs. Lines 783−784.

RESPONSE

Questions for Authors

Comments 1: Were root colonization levels consistent across plant species, or was there evidence of host preference (e.g., P. sativa vs. A. odoratum)?

Response 1: Based on our data, reported in the Figure 4 of the manuscript, it is not currently possible to firmly establish a particular association regarding host preference. Nevertheless, P. sativa and A. odoratum showed the highest percentages of root colonization, although in these plants the fungus may act differently, either as a pathogen or as an endophyte.

Comments 2: Beyond Poa trivialis, do the VOC effects extend to other grasses dominant in GB zones (e.g., Dactylis glomerata, Festuca rubra)?

Response 2: We tested the effects of VOCs under in vitro conditions only on P. trivialis, as it was the most abundant species in the fairy ring from which strain Calgam12 originated. The results of the VOC test are reported in Supplementary Figure S4. Nevertheless, we assume that these effects could also be extended to other species forming the greener belt.

Comments 3: Do you consider C. gambosa to act primarily as a necrotroph, or could it initially establish as a biotroph/endophyte before switching to pathogenic behavior?

Response 3: We think that C. gambosa has a broad trophic versatility: depending on its life stage, it can live as a endophyte, pathogen or saprotroph. The mycelium of C. gambosa might initially persist as a latent endophyte in plants and then switch to a pathogenic phase, extending into the soil as saprotroph, forming the fairy ring, where its activity reaches a maximum level at the fungal front. The severity of infection may depend on the plant species. These considerations are explained in the manuscript at the paragraph:” Direct pathogenicity of C. gambosa mycelium on plants in fungal front” and in the Figure 7. In other experiments, not included in this manuscript, as part of new study, we observed that plant colonization is systemic. The possibility of whole-plant colonization could also allow vertical transmission through seeds.

Comments 4: Could soil nutrient conditions (e.g., nitrogen enrichment) modulate the balance between pathogenic suppression at the fungal front and growth promotion behind it?

Response 4: We think that soil nutrient conditions may play this role by counteracting the antagonistic effects of the fungus and promoting plant growth, as we reported in the Discussion. Lines 474−479.

Comments 5: Were soil physicochemical properties (pH, N, P, K) considered in explaining NMDS community patterns, or are these attributed solely to fungal activity?

Response 5: We did not specifically examine these parameters, since the three fairy rings studied were located in the same grassland and close enough to each other to consider the soil physicochemical properties constant (see Figure 8). However, we assumed that the NMDS results may also be related to changes in soil physicochemical properties caused by fungal activity, as reported in the Discussion. Lines 342−351.

Comments 6: Do you hypothesize that plant colonization is essential for triggering C. gambosa fruiting, as suggested in the conclusions?

Response 6: We conducted numerous in vitro experiments attempting to induce C. gambosa fructification in the absence of plants, but without success. There are certainly many factors that could influence C. gambosa fructification in the field, but we believe that during the life cycle of the mycelium, which may be longer than expected, plant colonization plays an essential role in the fruiting process. For this reason, we conclude that this result could be important for cultivation, as we noted in the Introduction and Conclusion. Lines 112−114, 784−787.

Reviewer 3 Report

Comments and Suggestions for Authors

This study tested the plant-Fungus Interactions in Calocybe gambosa Fairy Rings. The manuscript is in general well-written, and the research topic is intriguing, the conclusions are supported by the results. However, after carefully reading the manuscript, I think the manuscript should be improved before publication. My detailed comments are as below.

  1. One limitation of this study is that the varying environmental conditions at the three selected sites may have impacted the results. Such as the temperature, air humidity, and soil properties. This paper does not consider the impact of these factors on the results.
  2. There are several formatting errors in the MS. For instance:

In line 100, please delete the duplicated "47".

In line 503, please revise the cited reference.

For Figure 5 and Figure S4, please correct the misspelling "lenght" to "length".

Please carefully check the entire MS.

  1. The seeds were sealed for 2 weeks to conduct germination and growth treatment in an experiment testing the effects of VOCs. Right? Such a long period of sealing will inherently be detrimental to plant growth, and this will affect the experimental results.

Author Response

Reviewer 3

This study tested the plant-Fungus Interactions in Calocybe gambosa Fairy Rings. The manuscript is in general well-written, and the research topic is intriguing, the conclusions are supported by the results. However, after carefully reading the manuscript, I think the manuscript should be improved before publication. My detailed comments are as below.

  1. One limitation of this study is that the varying environmental conditions at the three selected sites may have impacted the results. Such as the temperature, air humidity, and soil properties. This paper does not consider the impact of these factors on the results.
  2. There are several formatting errors in the MS. For instance:

In line 100, please delete the duplicated "47".

In line 503, please revise the cited reference.

For Figure 5 and Figure S4, please correct the misspelling "lenght" to "length".

Please carefully check the entire MS.

3. The seeds were sealed for 2 weeks to conduct germination and growth treatment in an experiment testing the effects of VOCs. Right? Such a long period of sealing will inherently be detrimental to plant growth, and this will affect the experimental results.

RESPONSE

Comments 1: One limitation of this study is that the varying environmental conditions at the three selected sites may have impacted the results. Such as the temperature, air humidity, and soil properties. This paper does not consider the impact of these factors on the results.

Response 1: Thank you for your suggestions. We did not consider environmental conditions such as temperature, air humidity, and soil properties because the three fairy rings studied were located in the same grassland and close enough to each other to assume that they developed under the same environmental conditions. See Figure 8.

Comments 2: There are several formatting errors in the MS. For instance:

In line 100, please delete the duplicated "47".

In line 503, please revise the cited reference.

For Figure 5 and Figure S4, please correct the misspelling "lenght" to "length".

Please carefully check the entire MS.

Response 2: Thank you for your suggestions.  We have corrected the formatting errors and the misspelling “length” in all the manuscript and Supplementary Files.

Comments 3: The seeds were sealed for 2 weeks to conduct germination and growth treatment in an experiment testing the effects of VOCs. Right? Such a long period of sealing will inherently be detrimental to plant growth, and this will affect the experimental results.

Response 3: Thank you for your suggestions. Even though the experimental conditions could have slightly affected plant growth, the control and test plates were subjected to the same growing time and conditions. Therefore, any secondary effects should have applied to all plates equally, allowing the results to be considered reliable. See Supplementary Figure S4.

Round 2

Reviewer 1 Report

Comments and Suggestions for Authors

Authors have positively answered my questions in the revision.

Reviewer 3 Report

Comments and Suggestions for Authors

I am glad that all my comments have been addressed. I think it‘s ready to be published.